# Olaf-World: Orienting Latent Actions for Video World Modeling

**Yuxin Jiang** [1 2]   **Yuchao Gu** [1]   **Ivor W. Tsang** [2]   **Mike Zheng Shou** [1]

https://showlab.github.io/Olaf-World

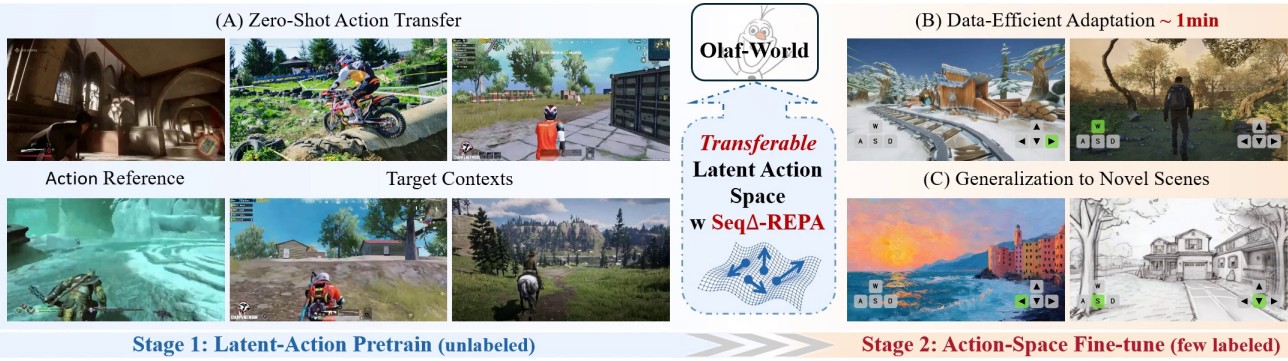

*Figure 1.* We present **Olaf-World**, an adaptable video world model pretrained with transferable latent actions learned via **SeqΔ-REPA**. Olaf-World enables (A) context-invariant zero-shot action transfer, (B) efficient adaptation to new action spaces with minimal labeled data (*e.g.*, 1 minute), and (C) improved generalization to novel scenes. Video results are available on the project page.

## Abstract

Scaling action-controllable world models is limited by the scarcity of action labels. While latent action learning promises to extract control interfaces from unlabeled video, learned latents often fail to transfer across contexts: they entangle scene-specific cues and lack a shared coordinate system. This occurs because standard objectives operate only *within* each clip, providing no mechanism to align action semantics across contexts. Our key insight is that although actions are unobserved, their *semantic effects* are observable and can serve as a shared reference. We introduce **SeqΔ-REPA**, a sequence-level control-effect alignment objective that anchors integrated latent action to temporal feature differences from a frozen, self-supervised video encoder. Building on this, we present **Olaf-World**, a pipeline that pretrains action-conditioned video world models from large-scale passive video. Extensive experiments demonstrate that our method learns a more structured latent action space, leading to stronger

zero-shot action transfer and more data-efficient adaptation to new control interfaces than state-of-the-art baselines.

## 1. Introduction

World models (Ha & Schmidhuber, 2018; Hafner et al., 2023; Parker-Holder et al.; Garrido et al., 2024; World Labs, 2025) that predict future observations under actions are essential for planning and interactive simulation. Recent video generative models (Brooks et al., 2024; Wan et al., 2025; Chen et al., 2025a; Kong et al., 2024; Peng et al., 2025; Gao et al., 2025b; Teng et al., 2025; Huang et al., 2025c; Gu et al., 2025) contain rich priors about visual and physical dynamics from internet-scale data, making them promising backbones for video world modeling. However, turning such models into action-controllable simulators still typically requires large-scale, frame-aligned action labels, which is costly and often tied to a specific domain or control interface (He et al., 2025; Sun et al., 2025; Yu et al., 2025a; Team et al., 2026).

Latent action learning (Edwards et al., 2019; Rybkin et al., 2019; Schmidt & Jiang, 2024; Ye et al., 2025) offers a scalable solution by discovering an action space directly from unlabeled videos: an inverse-dynamics encoder infers latent actions $z_i$ from observed transitions $(x_i, x_{i+1})$, and a forward model predicts future frames conditioned on past frames and inferred actions. Yet learning *transferable* latent actions remains challenging. Actions are considered

[1]Show Lab, National University of Singapore [2]CFAR & IHPC, Agency for Science, Technology and Research (A*STAR), Singapore. Correspondence to: Mike Zheng Shou <mike.zheng.shou@gmail.com>.

*Proceedings of the 43rd International Conference on Machine Learning*, Seoul, South Korea. PMLR 306, 2026. Copyright 2026 by the author(s).

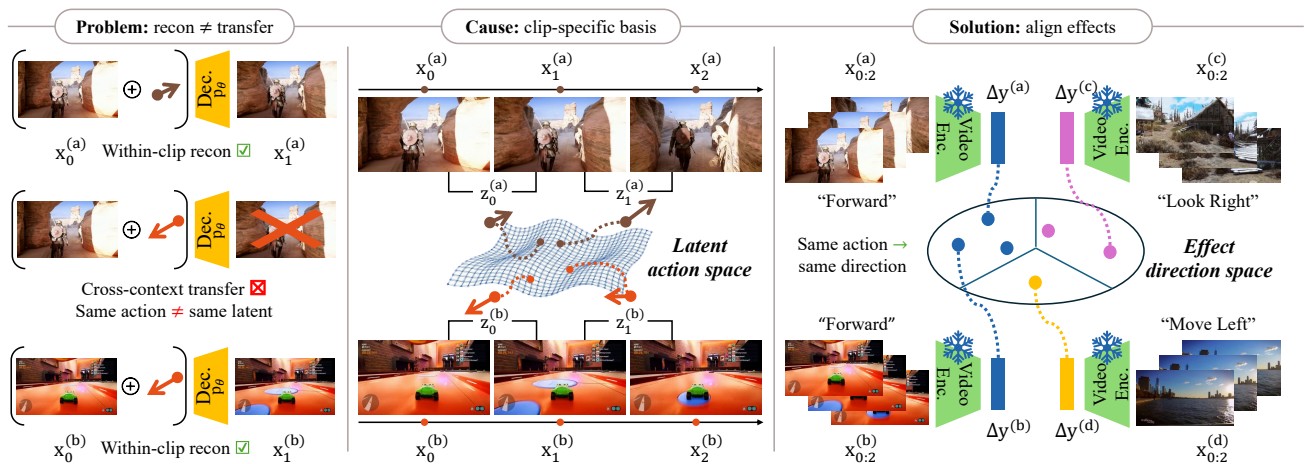

*Figure 2.* **Latent action learning. Problem**: transition-based latent action models (LAMs) can reconstruct well, but fail to transfer (the same semantic action, *e.g.*, "Forward", maps to different latent directions across contexts). **Cause:** the latent space is identified only up to a clip-specific basis, so there is no shared coordinate system. **Solution:** Seq$\Delta$-REPA uses the observable effect direction $\Delta y$ from a frozen video encoder as a shared reference and aligns latent actions to it, yielding consistent action semantics across contexts.

transferable if they preserve *control semantics* across contexts: transitions corresponding to the same underlying action should produce similar $z_i$ even when the visual context (appearance, viewpoint, layout, lighting, etc.) varies.

We identify two failure modes. First, inverse dynamics encoders often suffer from *shortcut learning* (Yang et al., 2025; Garrido et al., 2026): $z_i$ may rely on context-dependent visual cues rather than the underlying controllable cause, entangling the learned actions with scene appearance. Second, and more fundamentally, local reconstruction objectives are *non-identifiable across contexts* (Locatello et al., 2019; Khemakhem et al., 2020; Wang et al., 2023b). Because training is isolated to individual clips, the model is not encouraged to use a shared latent coordinate system across contexts, so the same semantic action (*e.g.*, "move forward") can correspond to different latent directions in different environments (see Figure 2, Left). Together, these issues prevent a shared control interface from emerging: identical action semantics need not map to a consistent region of latent space, undermining transfer and downstream controllability.

To address these, we propose **Seq$\Delta$-REPA**, a sequence-level objective that regularizes the latent space via **control-effect alignment**. Our key insight is that while explicit action labels are unavailable, the *semantic effect* of control is observable in video: transitions driven by similar underlying actions should induce similar *semantic change* across contexts, despite appearance differences. We formalize this by leveraging a frozen, self-supervised video encoder (Tong et al., 2022; Assran et al., 2025) to define a target *effect direction* based on the net semantic change of a short clip (see Figure 2, Right). Crucially, temporal feature differences naturally suppress spatial details and emphasize dynamics, making the reference stable under context shifts.

Seq$\Delta$-REPA then aligns the *integrated* latent action inferred over the same window to this effect direction. This provides a shared global reference that encourages consistent action meanings across contexts and discourages reliance on context-specific visual shortcuts.

Using the latent actions learned with Seq$\Delta$-REPA as a consistent control interface, we present **Olaf-World**, a pipeline for pretraining action-conditioned video world models on large-scale passive video. Thanks to the structural alignment of our representation, it fundamentally improves the capabilities of the downstream world model (see Figure 1): (i) Context-invariant zero-shot action transfer: latent actions extracted from demonstrations in one context can be reused to induce similar control effects in new contexts. (ii) Efficient adaptation: when true labels are available, we learn a lightweight mapping to our pretrained action space, enabling adaptation with minimal data and parameter updates. (iii) Better generalization to unseen context: because latent-action pretraining exposes the model to diverse transitions, Olaf-World generalizes better to novel scenes than models trained from scratch on labeled datasets. In summary, our key contributions are as follows:

- We **characterize** cross-context non-identifiability in latent action learning, showing why step-wise reconstruction fails to learn transferable control.

- We propose **Seq$\Delta$-REPA**, a novel sequence-level control-to-effect alignment objective that anchors latent action trajectory to semantic change derived from self-supervised video representations, encouraging context-invariant action semantics.

- We introduce **Olaf-World**, a pretraining pipeline that learns action-controllable video world models from

passive video, enabling reliable cross-context action transfer and efficient adaptation with minimal labeled data.

## 2. Related Work

### 2.1. Learning Latent Action from Videos

Latent action models aim to infer latent controls from unlabeled video. They have been used either as (i) unified control interfaces for interactive world models (Bruce et al., 2024; Gao et al., 2025a; Jang et al., 2025), or as (ii) action representations for policy learning, specifically to bridge cross-embodiment gaps in robotics (Ye et al., 2025; Bu et al., 2025; Kim et al., 2025; Chen et al., 2025b;c; Yang et al., 2025) and (iii) enable observation-only offline RL (Schmidt & Jiang, 2024; Nikulin et al., 2025). Most LAMs learn an inverse model that infers per-step latents from observed transitions and a forward decoder trained with reconstruction or prediction objectives. Both discrete (VQ-based) (Schmidt & Jiang, 2024; Bruce et al., 2024; Ye et al., 2025) and continuous latent (Gao et al., 2025a; Yang et al., 2025; Garrido et al., 2026) parameterizations have been explored. Prior work recognizes that local transition-based objectives are sensitive to nuisance factors and action-correlated distractors, which can induce shortcut solutions and degrade downstream use (Nikulin et al., 2025; Bu et al., 2025; Garrido et al., 2026). To mitigate this, existing methods impose latent space constraints (Gao et al., 2025a; Garrido et al., 2026) or design objectives that emphasize motion over pixel appearance (Chen et al., 2025c; Bu et al., 2025; Yang et al., 2025; Bi et al., 2025). However, these methods operate on isolated clips and do not, by themselves, enforce that latent action semantics remain consistent across environments. Seq$\Delta$-REPA fixes this by anchoring latent actions to a global effect reference via sequence-level alignment.

### 2.2. Video World Model

World models predict future observations and support planning or interactive simulation in domains such as games, robotics, and driving (Parker-Holder et al.; Agarwal et al., 2025; Gao et al., 2024; Bar et al., 2025). Most action-controllable video world models rely on *explicit* control signals collected from interactive game engines (*e.g.*, Unreal Engine, Minecraft), where frame-level keyboard/mouse inputs and other interaction annotations are logged as controls (Decart et al., 2024; Alonso et al., 2024; Valevski et al., 2025; Xiao et al., 2025b). This yields strong controllability, but also ties the learned model to a specific action schema and data-collection pipeline (He et al., 2025; Tang et al., 2025; Sun et al., 2025; Hong et al., 2025; Team et al., 2026; Ye et al., 2026). Latent-action world models instead infer a control interface directly from videos, enabling interaction without ground-truth actions (Bruce et al., 2024; Gao

et al., 2025a; Wang et al., 2025; Garrido et al., 2026). However, their controllability and transfer ultimately depend on whether the learned latent action space is consistent across contexts—precisely the bottleneck our work addresses.

### 2.3. Representation Alignment

Alignment methods match internal features of generative models to large self-supervised encoders to improve semantics fidelity and training efficiency. While initially focused on spatial features in image generation (Yu et al., 2025b; Leng et al., 2025; Singh et al., 2025), recent video extensions have incorporated temporal structure, aligning the internal states of video generators to those of pretrained video encoders (Zhang et al., 2025; Chefer et al., 2025; Bhowmik et al., 2025). These approaches primarily aim to improve the generator's internal *state* representations for higher-quality synthesis, *i.e.*, feature-to-feature alignment. In contrast, we use the pretrained spatiotemporal encoder (Tong et al., 2022; Assran et al., 2025) as a reference to supervise latent actions by matching semantic effects (feature change), *i.e.*, a control-to-effect alignment.

## 3. Method

Our goal is to learn an action-controllable video world model from unlabeled video. We formulate the proposed Olaf-World into two stages: (1) learning a *transferable* latent action space that disentangles dynamics from visual context (Section 3.1), and (2) training a video generative world model conditioned on these latent actions (Section 3.2).

### 3.1. Latent Action Model

$\beta$-**VAE.** Given a clip $x_{0:K}$, we model each transition $(x_i, x_{i+1})$ with a latent action $z_i \in \mathbb{R}^{d_z}$ for $i = 0, \ldots, K-1$. A standard latent action model (Schmidt & Jiang, 2024; Gao et al., 2025a) consists of a causal inverse-dynamics encoder that produces $q_\phi(z_i \mid x_{0:i+1})$ and a forward decoder that predicts the next frame $p_\theta(x_{i+1} \mid x_i, z_i)$, to ensure the latent captures the dynamics required to explain the pixel shift. The model is trained with the step-wise $\beta$-VAE objective (Higgins et al., 2017; Alemi et al., 2017):

$$\mathcal{L}_{\theta,\phi}^{\text{VAE}} = \frac{1}{K} \sum_{i=0}^{K-1} \Big( - \mathbb{E}_{q_\phi(z_i|x_{0:i+1})} \big[ \log p_\theta(x_{i+1} \mid x_i, z_i) \big]$$
$$+ \beta \, \text{KL}(q_\phi(z_i \mid x_{0:i+1}) \, \| \, p(z_i)) \Big), \tag{1}$$

where $p(z_i)$ is a fixed prior $\mathcal{N}(0, I)$. While Eq. (1) can achieve low one-step prediction error, it does not, on its own, ensure a semantically consistent action space across contexts. We summarize two failure modes: **(i) Shortcut learning (context leakage):** Because the posterior condi-

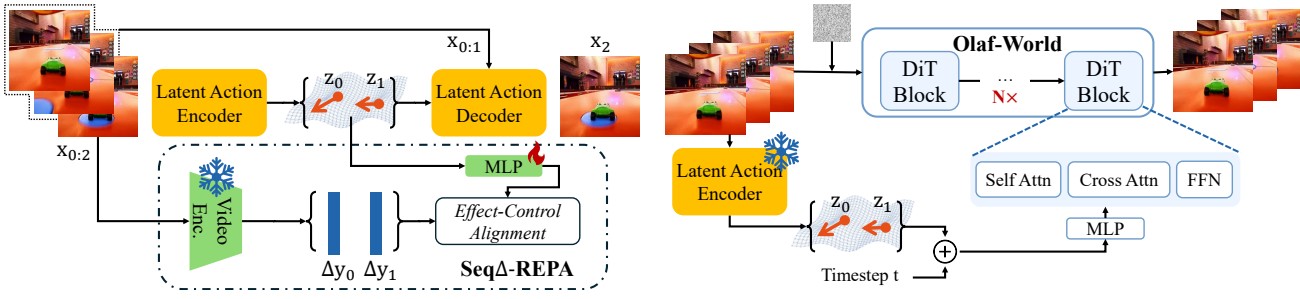

*(a)* **Seq$\Delta$-REPA latent action learning.**    *(b)* **Olaf-World action-aware pretraining.**

*Figure 3.* **Overall pipeline.** (a) We train a latent action model (LAM) and encourage cross-context consistency by aligning *action effects* in a frozen video-feature space using Seq$\Delta$-REPA. (b) We then apply the frozen LAM to unlabeled videos to extract latent-action sequences, and use them as a unified control interface to pretrain an action-conditioned video world model.

tions on $x_{i+1}$, *i.e.*, $q_\phi(z_i \mid x_{0:i+1})$, an expressive decoder can reduce the loss by encoding context-dependent cues correlated with $x_{i+1}$ rather than a transferable control into $z_i$. **(ii) Cross-context non-identifiability:** Since the loss never compares latents across trajectories, the latent coordinate system is unconstrained and may drift across contexts: the same semantic motion may map to different directions of latent space in different videos, breaking transfer (see Appendix A for a formal discussion).

**Seq$\Delta$-REPA.** To resolve these ambiguities, we introduce a sequence-level alignment constraint that anchors latent actions to an effect signal that is comparable across videos and contexts (see Figure 3a). Let $f$ be a frozen self-supervised video encoder (*e.g.*, V-JEPA2 ViT (Assran et al., 2025)). Given $x_{0:K}$, it outputs spatial-temporal visual tokens and we spatially pool to obtain per-frame descriptors $s_i \in \mathbb{R}^D$. We define the clip's *effect direction* as the net direction of feature change:

$$\tau_* = \frac{1}{K} \sum_{i=0}^{K-1} \left( s_{i+1} - s_i \right) \in \mathbb{R}^D. \quad (2)$$

Because $\tau_*$ is computed from temporal differences and averaged over time, it emphasizes coherent temporal change in feature space and $\Delta s$ is less sensitive to static appearance.

On the latent-action side, the inverse model infers a sequence of latent actions $z_{0:K-1}$. We aggregate them and map them into the encoder feature space:

$$\bar{z} = \frac{1}{K} \sum_{i=0}^{K-1} z_i \in \mathbb{R}^{d_z}, \qquad u = h_\psi(\bar{z}) \in \mathbb{R}^D, \quad (3)$$

where $h_\psi : \mathbb{R}^{d_z} \to \mathbb{R}^D$ is a trainable MLP projection head.

We then align the integrated control direction $u$ with the effect direction $\tau_*$ using cosine similarity:

$$\mathcal{L}_\psi^{\text{Seq}\Delta\text{-REPA}} = 1 - \langle \text{norm}(u), \text{norm}(\tau_*) \rangle. \quad (4)$$

Unlike feature-to-feature alignment, Eq. (4) imposes a *control-to-effect* constraint: it aligns integrated latent con-

trol to a shared notion of semantic change, encouraging consistent action meaning across contexts.

**Final training objective.** We train $(\theta, \phi, \psi)$ with:

$$\mathcal{L}_{\text{LAM}} = \mathcal{L}_{\theta,\phi}^{\text{VAE}} + \lambda \mathcal{L}_\psi^{\text{Seq}\Delta\text{-REPA}}, \quad (5)$$

while keeping the reference encoder $f$ frozen and $\lambda > 0$ is the loss weight.

### 3.2. Olaf-World

**Action-aware Pretraining.** Given a video $x_{0:T}$, the frozen LAM generates per-frame latent actions $z_{0:T-1} \in \mathbb{R}^{d_z}$. We build on a pretrained latent image-to-video diffusion transformer (DiT) (Peebles & Xie, 2023; Chen et al., 2025a) and train on sequences of frames paired with latent actions using the standard flow-matching objective (Liu et al., 2023) (see Figure 3b). Per-frame $z_t$ is linearly projected and added to the diffusion timestep embedding. The fused embedding is then mapped to per-block AdaLN-Zero modulation parameters that condition each DiT block (Peebles & Xie, 2023). Because the backbone operates on latents encoded by a 3D video VAE, the input video is temporally compressed by a factor of $r{=}4$ (Wan et al., 2025; Chen et al., 2025a). Accordingly, we group every $r$ consecutive per-step actions into one latent-time conditioning vector, following Yu et al. (2025a). As a result, the world model is conditioned on LAM latents, providing a unified control interface that transfers across environments with different raw action conventions.

**Specific-world Adaptation.** In a target interactive environment, we observe explicit actions $a_t$ from an environment-specific action space. We learn a small action adapter $A_\eta$ that maps environment actions to latent actions: $\hat{z}_t = A_\eta(a_t)$, and control the pretrained world model using $\hat{z}_t$. For a discrete action set $\mathcal{A}$, $A_\eta$ can be implemented as an embedding table $E \in \mathbb{R}^{|\mathcal{A}| \times d_z}$ with $\hat{z}_t = E[a_t]$. We initialize $E$ with class-wise prototypes computed from the target data: for each action $a \in \mathcal{A}$, we run the frozen LAM

on segments labeled with $a$ and set $E[a]$ to the average inferred latent action. We then finetune (i) the action adapter and (ii) a small LoRA on the backbone using the same flow-matching objective. This quickly specializes the model to new action spaces while preserving the globally aligned latent control semantics learned from passive video.

## 4. Experiments

We validate the performance of Seq$\Delta$-REPA and the effect of the better latent actions to downstream applications through extensive experiments. In particular, we investigate the following questions:

- **RQ1 (Structure):** Do learned latents encode action semantics that are linearly decodable and consistent across domains? (Section 4.2)

- **RQ2 (Transfer):** Does this alignment enable zero-shot control transfer to new context? (Section 4.3)

- **RQ3 (Adaptation):** Does the aligned latent space enable data-efficient adaptation to specific control interfaces? (Section 4.4)

### 4.1. Experimental Setup

**Dataset.** We train the latent action model and the action-conditioned world model on the 3D Rendering and City Walking categories of MiraData (Ju et al., 2024). For specific-world adaptation and controlled evaluation, we use MIND (Ye et al., 2026), an open-domain dataset with frame-aligned action labels collected in Unreal Engine 5. MIND contains two disjoint subsets with different scenes and camera rigs: First-Person (1ST-P) and Third-Person (3RD-P). Both share the same 8-action label space: navigation (W/S/A/D: forward/back/left/right) and camera control ($\uparrow$/$\downarrow$/$\leftarrow$/$\rightarrow$: look up/down/left/right). This split allows us to rigorously test cross-context transfer under both appearance shifts and viewpoint shifts.

**Implementation Details.** Our latent action encoder is a spatiotemporal Transformer with causal temporal attention and latent dimension $d_z=32$. It is trained with window size $K=16$ and $\lambda=0.02$. Our world model is built on the SkyReels-V2-1.3B I2V DiT backbone (Chen et al., 2025a), trained at 540p on clips of $T=97$ frames (25 latent frames). All experiments are conducted on NVIDIA H200 GPUs. Additional implementation details are provided in Appendix B.

**Baselines.** We compare against AdaWorld (Gao et al., 2025a), a state-of-the-art latent-action world model. For a controlled comparison, we run AdaWorld with the same video-model backbone, data, and training/adaptation budgets as ours, while keeping its official latent-action training

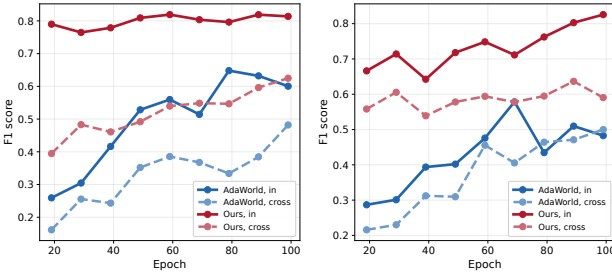

*(a)* Probe: 1st$\to$ {1st, 3rd}.  *(b)* Probe: 3rd$\to$ {3rd, 1st}.

*Figure 4.* **In-/cross-domain linear probing over training.** Solid: in-domain; dashed: cross-domain evaluation (1st-P$\leftrightarrow$3rd-P). Seq$\Delta$-REPA consistently improves both in-domain and cross-domain probe performance.

*Table 1.* **In-/cross-domain linear probing** (Macro-F1, $\uparrow$). Gray columns denote cross-domain probing (source$\to$target).

| Method | 1st$\to$1st | 1st$\to$3rd | 3rd$\to$3rd | 3rd$\to$1st |
|---|---|---|---|---|
| AdaWorld | 0.6004 | 0.4820 | 0.4827 | 0.4999 |
| Ours | **0.8138** | **0.6250** | **0.8256** | **0.5904** |

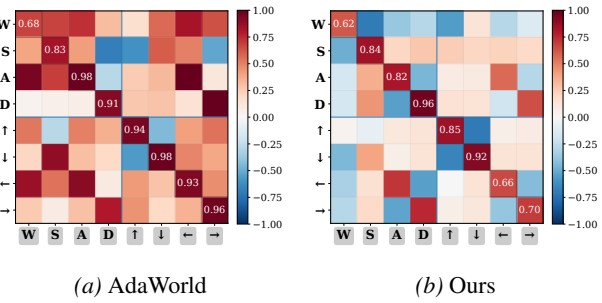

*(a)* AdaWorld  *(b)* Ours

*Figure 5.* **Cross-domain action similarity.** Cosine similarity between per-action prototypes from 1ST-P (rows) and 3RD-P (columns). Seq$\Delta$-REPA produces a more diagonal-dominant matrix (stronger one-to-one matching across contexts).

pipeline and configuration unchanged. Thus, differences isolate the effect of the latent-action learning objective.

**Evaluation.** We assess latent-action structure (probing + prototype similarity), transfer (action-sequence transfer), and adaptation (VBench (Huang et al., 2024) + RPE (Hong et al., 2025)). Protocols and metric details are given per section, with full implementation in Appendix C.

### 4.2. Latent Space Diagnostics

#### 4.2.1. CROSS-CONTEXT LINEAR PROBING

**Setup.** We evaluate whether the learned latent action space $z_t$ is linearly separable and invariant to domain shifts. Following Zhang et al. (2022), we train a linear probe to predict the 8 atomic actions from $z_t$ for each checkpoint. To test *context invariance*, we use a **cross-domain probing protocol**: we train and validate the probe on 1ST-P, select the

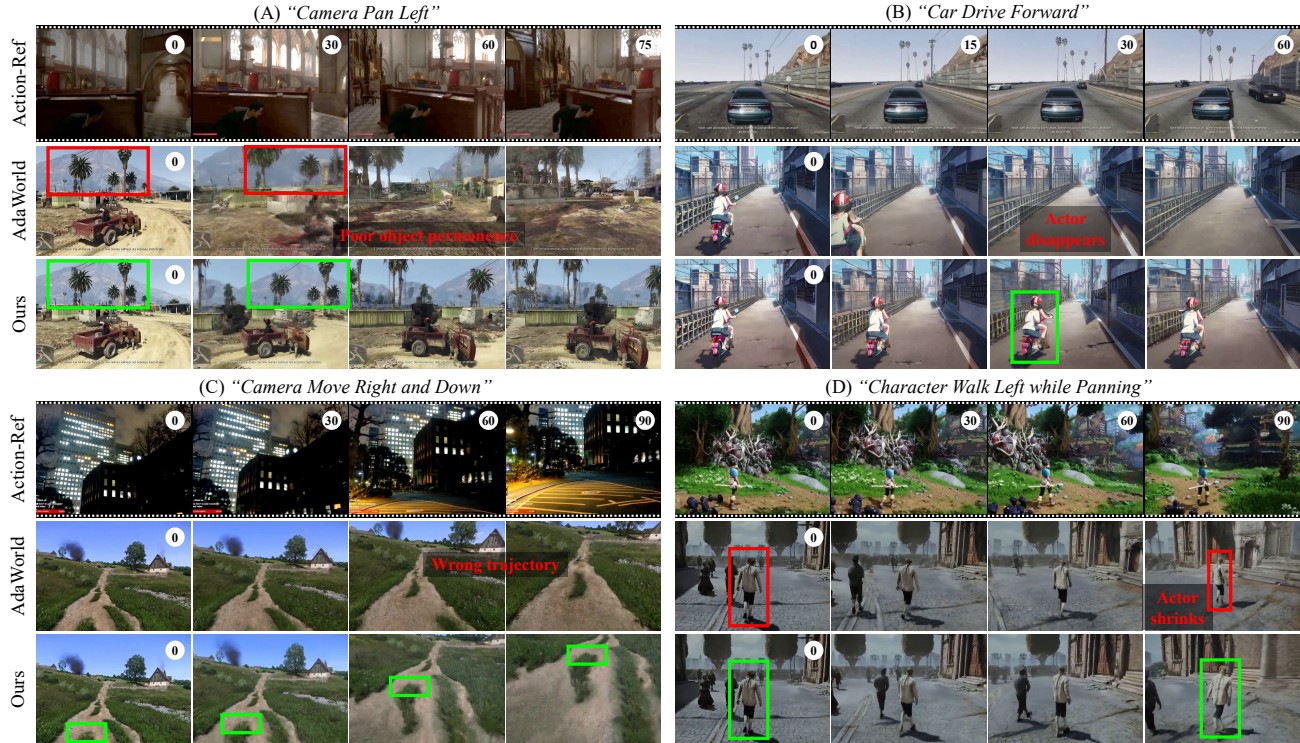

Figure 6. **Zero-shot action-sequence transfer.** We extract an action sequence from a reference clip (top) and apply it zero-shot to a different target context. AdaWorld often shows temporal wash-out, agent drop-out, and motion drift, whereas Olaf-World performs better in target appearance preservation and motion faithfulness. Numbers denote frame indices. See the project page for video comparisons.

best checkpoint by in-domain validation F1 score, and then evaluate the same probe zero-shot on 3RD-P. We repeat the reverse direction (3RD-P→1ST-P). We report Macro-F1 for class-balanced comparison.

**Results.** Figure 4 shows that SeqΔ-REPA learns latent actions that are both more linearly decodable and more context-invariant. Across checkpoints, Olaf-World achieves higher in-domain Macro-F1 and consistently outperforms AdaWorld under cross-domain evaluation in both directions (1ST-P↔3RD-P), indicating improved alignment of action semantics across viewpoint and appearance shifts. Notably, gains are largest when probing on the more challenging 3RD-P subset, where AdaWorld saturates at low Macro-F1 while ours remains substantially higher (Table 1). Moreover, by aligning effect directions, our method bootstraps early-stage learning and reduces training-time fluctuations, yielding a more stable and transferable latent structure.

### 4.2.2. CROSS-CONTEXT ACTION CONSISTENCY

**Setup.** To test whether action semantics are consistent across context, we compute an action prototype (class centroid) separately within each domain and visualize the cross-domain cosine similarity between prototypes (1ST-P as rows, 3RD-P as columns). A well-aligned latent space should be diagonal-dominant, i.e., each 1ST-P action is most similar to its 3RD-P counterpart.

**Results.** Figure 5 compares cross-context prototype similarity between 1ST-P and 3RD-P. The Adaworld baseline (Figure 5a) shows high similarity everywhere, meaning different actions in 1ST-P often look "similar" to multiple actions in 3RD-P. This indicates the latent space is not uniquely action-specific under context shift, i.e., weak cross-context identifiability. In contrast, Ours (Figure 5b) is visibly more contrastive: matching actions retain high similarity, while non-matching pairs are pushed closer to zero or negative. This suggests SeqΔ-REPA learns more consistent, transferable action semantics across viewpoint and appearance changes. The remaining confusion appears in yaw "look" actions (← / →), which are expected to be less aligned because the same control actually induces different observable motion under egocentric vs. third-person camera rigs.

### 4.3. Zero-shot Action Transfer

**Setup.** We qualitatively evaluate whether the model follows the control signal $z_t$ independently of visual context. We extract a latent action sequence $z_{0:T}$ from a reference clip and use it to drive generation from a different target initial frame. Successful transfer requires reproducing the reference *motion* while preserving the target *appearance*.

**Results.** Figure 6 illustrates that Olaf-World transfers action

*Table 2.* **Adapting world models to target control domains with different amounts of labeled data (#Adapt Videos).** We report VBench visual metrics (↑) and action accuracy via RPE (↓). Olaf-World achieves the lowest RPE in all settings, indicating the most faithful action following. Best per domain and budget is in **bold**.

| Method | # Adapt Videos | 1ST-P | | | | 3RD-P | | | |
|---|---|---|---|---|---|---|---|---|---|
| | | Visual Quality | | Action Accuracy (RPE) | | Visual Quality | | Action Accuracy (RPE) | |
| | | Image Qual. ↑ | Temp. Cons. ↑ | Trans ↓ | Rot. ↓ | Image Qual. ↑ | Temp. Cons. ↑ | Trans ↓ | Rot. ↓ |
| DirectAct | 0 | **0.7213** | 0.8993 | 0.0703 | 1.4311 | **0.6970** | 0.9086 | 0.0897 | 0.7968 |
| AdaWorld | | 0.5600 | **0.9226** | 0.0470 | 1.0844 | 0.6102 | **0.9344** | 0.0723 | 0.6067 |
| Ours | | 0.5400 | 0.9123 | **0.0387** | **0.8773** | 0.5909 | 0.9203 | **0.0461** | **0.4873** |
| DirectAct | 1 | 0.5269 | 0.8828 | 0.0672 | 1.2822 | 0.6019 | 0.8851 | 0.0708 | 0.8543 |
| AdaWorld | | 0.5623 | 0.8955 | 0.0318 | 0.6420 | **0.6033** | **0.8989** | 0.0525 | 0.4659 |
| Ours | | **0.5726** | **0.9015** | **0.0284** | **0.4680** | 0.5844 | 0.8974 | **0.0348** | **0.3861** |
| DirectAct | 50 | 0.5936 | **0.9345** | 0.0351 | 0.4527 | 0.6265 | 0.9286 | 0.0402 | 0.3846 |
| AdaWorld | | 0.6177 | 0.9239 | 0.0263 | 0.3834 | 0.6459 | **0.9306** | 0.0393 | 0.3353 |
| Ours | | **0.6312** | 0.9263 | **0.0230** | **0.3785** | **0.6486** | 0.9287 | **0.0222** | **0.2082** |

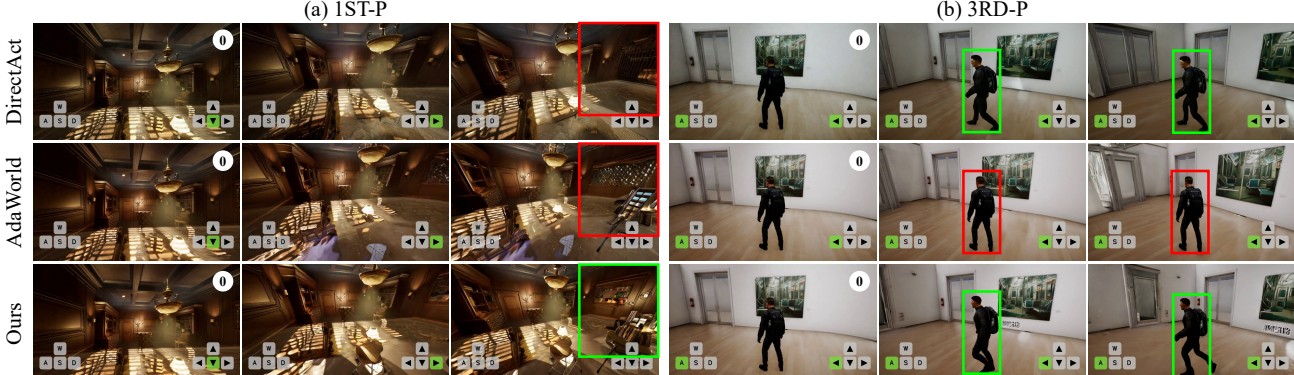

(a) 1ST-P  (b) 3RD-P

*Figure 7.* **Qualitative comparison of action-conditioned generation after adaptation.** Given the same initial frame and action sequence, Olaf-World follows controls more faithfully and preserves appearance consistency as new regions are revealed. Actions are transition-aligned: $a_t$ corresponds to the change from $x_t$ to $x_{t+1}$. Zoom in for details.

sequences more reliably while better preserving the target context. Across all four cases, AdaWorld often degrades under transfer: it shows (i) temporal wash-out and instability (A), (ii) loss of the controlled character or scale drift (B,D), and (iii) trajectory drift toward generic motions that deviate from the reference (C). In contrast, Olaf-World maintains scene and subject persistence while faithfully executing the intended motion. Overall, these results indicate that Seq$\Delta$-REPA produces a control signal whose semantics remain action-specific under large context shifts. Additional examples are provided in Appendix D.

### 4.4. World Model Adaptation

#### 4.4.1. DATA-EFFICIENT ADAPTATION

**Setup.** We study how efficiently a pretrained video world model can be adapted to a target control interface under limited labeled interaction. We compare: (a) **DirectAct**, conditioning directly on ground-truth actions; (b) **Ada-World**, latent-action pretraining with vanilla $\beta$-VAE; (c) **Olaf-World**, latent-action pretraining with $\beta$-VAE + Seq$\Delta$-REPA. All methods use the same video backbone and adap-

tation capacity (LoRA rank 16 with matched steps and optimizer). We vary the labeled adaptation set size (#Adapt Videos $\in \{0, 1, 50\}$; $\approx 0$, 1 minute, and 2 hours). We measure video quality using VBench (Huang et al., 2024) and controllability with translational and rotational relative pose error (RPE) following Hong et al. (2025).

**Quantitative results.** Table 2 shows that Olaf-World achieves the lowest RPE-trans and RPE-rot across all adaptation budgets on both 1ST-P and 3RD-P, indicating the most faithful action following. Compared to AdaWorld, Olaf-World consistently improves controllability while keeping comparable video quality, suggesting that Seq$\Delta$-REPA learns a latent control representation that is easier to adapt. DirectAct with 0 videos reduces to standard image-to-video generation, explaining its strong visual scores but uninformative controllability. With action supervision, DirectAct improves, but remains less controllable than latent-action pretraining under the same rank-16 LoRA setting. We expect the gap to narrow with larger adaptation capacity (*e.g.*, higher LoRA rank or full fine-tuning).

**Qualitative results.** Figure 7 is consistent with the quanti-

*Table 3.* **Generalization to unseen visual contexts after adaptation.** Olaf-World achieves the lowest RPE, indicating the most faithful action following under appearance shift.

| Model | Visual & Temporal Quality | | Action Accuracy (RPE) | |
|---|---|---|---|---|
| | Image Qual. ↑ | Temp. Cons. ↑ | Trans ↓ | Rot. ↓ |
| DirectAct | **0.6322** | 0.8585 | 0.0547 | 1.2343 |
| AdaWorld | 0.6181 | 0.8719 | 0.0482 | 1.7063 |
| Ours | 0.6274 | **0.8743** | **0.0478** | **1.2221** |

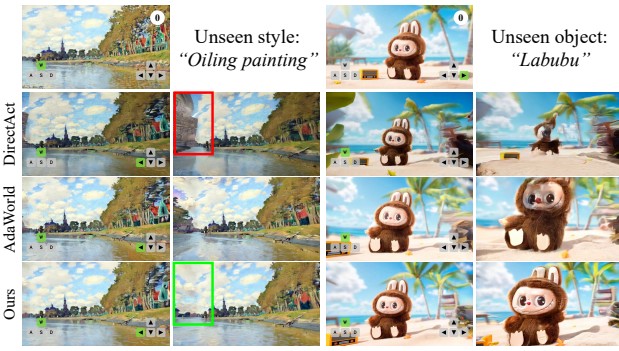

*Figure 8.* **Qualitative generalization under unseen contexts.** Left: baselines often break style consistency when completing newly revealed regions, while Right: baselines show subject drift, whereas Olaf-World better preserves a stable appearance under the same action sequence. Zoom in for details.

tative trends. After full adaptation (50 videos), Olaf-World follows the intended controls more reliably and keeps the generated world visually consistent: when the camera turns or the agent moves sideways, newly visible regions are synthesized with stable details that match the initial frame (Figure 7 (Left)). In contrast, AdaWorld is less reliable under multi-key controls (*e.g.*, 3RD-P turn+move-left): rollouts often rotate without the desired leftward motion, leading to less faithful action-conditioned generation.

### 4.4.2. GENERALIZATION TO UNSEEN CONTEXTS

**Setup.** We evaluate whether the adapted simulator remains reliable when exploring diverse visual worlds at test time. Using the fully adapted models from Section 4.4.1 (1ST-P action space), we construct an OOD test set of 50 initial frames spanning diverse styles and scenes. We report the same metrics.

**Quantitative results.** Table 3 shows that Olaf-World retains the best controllability under unseen visual contexts, achieving the lowest RPE. This suggests the learned latent control remains usable when the appearance shifts, rather than overfitting to the adaptation visuals.

**Qualitative results.** Figure 8 highlights two representative cases. In unseen styles, baselines often break style consistency when hallucinating newly revealed regions as the camera moves. In unseen objects, baselines struggle to keep object identity while making its pose/scale/viewpoint evolve

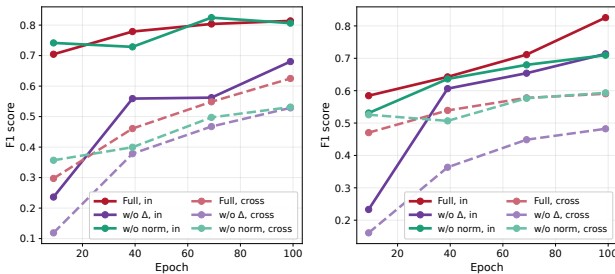

*(a)* Probe: 1st→ {1st, 3rd}.  *(b)* Probe: 3rd→ {3rd, 1st}.

*Figure 9.* **Ablations of Seq△-REPA on in-/cross-context linear probing.** Solid: in-domain; dashed: cross-domain evaluation.

*Table 4.* **Seq△-REPA ablations** (Macro-F1, ↑). Gray columns denote cross-domain probing (source→target).

| Method | 1st→1st | 1st→3rd | 3rd→3rd | 3rd→1st |
|---|---|---|---|---|
| w/o △ | 0.6805 | 0.5287 | 0.7137 | 0.4823 |
| w/o norm | 0.8064 | 0.5311 | 0.7096 | **0.5934** |
| Full | **0.8138** | **0.6250** | **0.8256** | 0.5904 |

in a way that matches the commanded actions. Olaf-World better preserves appearance consistency while producing action-consistent changes under the same action sequence. Overall, these results indicate that latent-action pretraining improves OOD robustness of action-conditioned dynamics.

### 4.5. Ablation Studies

#### 4.5.1. LATENT ACTION LEARNING

**Seq-△-REPA design.** We ablate key design choices in Seq△-REPA: (i) **w/o △**, which aligns static features rather than temporal effect directions; and (ii) **w/o norm**, which removes $\ell_2$ normalization and replaces cosine alignment with a scale-sensitive MSE loss. Figure 9 reports in-/cross-context linear probing under the same protocol as Section 4.2.1. Removing △ causes a clear drop in Macro-F1 (see Table 4), suggesting that aligning static features allows context-dependent spatial cues to leak into the action representation, so the probe becomes less separable and much less consistent across domains. Without normalization, the alignment becomes sensitive to feature magnitude, which can vary across domains. This destabilizes the learned latents, resulting in not reliably good across both domains. Overall, the full objective performs best and most consistently across domains, supporting the importance of aligning *action effects* with a stable, scale-invariant similarity.

**Reference encoder sensitivity.** We study the effect of the frozen self-supervised reference encoder $f$ used to compute target effect directions. Table 5 reports results under the same probing protocol. For efficiency, all models in this sensitivity study are trained with the same 10-epoch schedule. Despite this shorter schedule, clear trends emerge: video encoders, including V-JEPA2 (Assran et al., 2025) and Video-

### (a) Data budget sweep (fixed adaptation capacity)

| #Vids | Visual & Temporal Quality | | Action Accuracy (RPE) | |
|---|---|---|---|---|
| | Image Qual. ↑ | Temp. Cons. ↑ | Trans ↓ | Rot ↓ |
| 0 | 0.5400 | 0.9123 | 0.0387 | 0.8773 |
| 1 | 0.5726 | 0.9015 | 0.0284 | 0.4680 |
| 3 | **0.6542** | **0.9274** | 0.0304 | 0.4187 |
| 5 | 0.6171 | 0.9139 | 0.0284 | 0.4893 |
| 10 | 0.6311 | 0.9218 | 0.0271 | 0.4416 |
| 25 | 0.6321 | 0.9239 | 0.0250 | 0.3989 |
| 50 | 0.6312 | 0.9263 | **0.0230** | **0.3785** |

### (b) LoRA rank sweep (fixed data budget)

| Rank | Visual & Temporal Quality | | Action Accuracy (RPE) | |
|---|---|---|---|---|
| | Image Qual. ↑ | Temp. Cons. ↑ | Trans ↓ | Rot ↓ |
| 16 | 0.6312 | 0.9263 | 0.0230 | 0.3785 |
| 32 | 0.6265 | 0.9249 | 0.0230 | 0.3915 |
| 64 | **0.6394** | 0.9257 | 0.0251 | 0.3633 |
| 128 | 0.6309 | **0.9304** | 0.0213 | 0.3202 |
| 256 | 0.6372 | 0.9265 | 0.0220 | **0.2928** |
| Full | 0.6267 | 0.9210 | **0.0185** | 0.2980 |

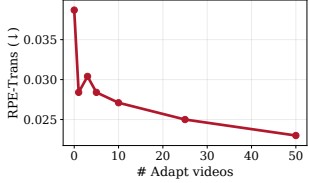 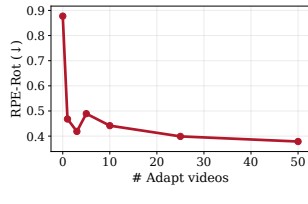 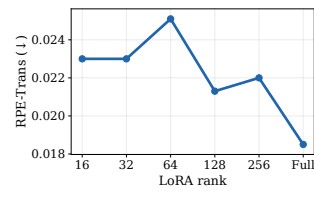 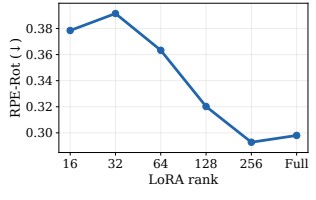

(c) RPE-Trans vs. #Videos     (d) RPE-Rot vs. #Videos     (e) RPE-Trans vs. Rank     (f) RPE-Rot vs. Rank

*Figure 10.* **World model adaptation scaling. Top:** quantitative results when varying the labeled target-domain data budget with LoRA rank fixed ($r$=16; left), and when varying LoRA rank with the data budget fixed to 50 videos (right). **Bottom:** corresponding scaling curves for action-following accuracy measured by RPE-Trans and RPE-Rot (lower is better). Across both sweeps, video quality remains comparable, while controllability improves with more labeled adaptation videos and larger adaptation capacity. **Bold** and underline denote the best and second-best results within each column, respectively.

*Table 5.* **Alignment to different visual reference encoders** (Macro-F1, ↑). Gray columns denote cross-domain probing (source→target). NONE denotes the AdaWorld baseline

| Ref. encoder | 1st→1st | 1st→3rd | 3rd→3rd | 3rd→1st |
|---|---|---|---|---|
| V-JEPA2 | **0.7044** | 0.2972 | 0.5845 | 0.4704 |
| VideoMAEv2 | 0.6626 | **0.4697** | **0.6616** | **0.5377** |
| DINOv3 | 0.2334 | 0.1782 | 0.2869 | 0.1910 |
| NONE | 0.1849 | 0.1382 | 0.2105 | 0.1593 |

MAEv2 (Wang et al., 2023a), substantially outperform the image-only encoder DINOv3 (Siméoni et al., 2025). This suggests that temporal modeling in the reference encoder is important for constructing transferable effect-direction targets. Both video encoders remain effective despite different pretraining objectives, indicating that Seq-Δ-REPA is not tied to a specific backbone but benefits from a frozen representation space with temporal consistency and sensitivity to video dynamics.

#### 4.5.2. WORLD MODEL ADAPTATION

**Data budget.** We study how downstream adaptation scales with labeled target-domain supervision. Beyond the main-experiment budgets $\{0, 1, 50\}$, we evaluate $\{0, 1, 3, 5, 10, 25, 50\}$ labeled videos, corresponding to approximately $\{0, 1, 6, 13, 26, 60, 120\}$ minutes of supervision. Here, the 0-video setting denotes zero-shot transfer without target-domain action labels. Table 10a shows that action accuracy improves substantially with more supervision, with the steepest gains in the low-data regime. Video quality remains comparable, suggesting that additional la-

bels mainly improve control alignment rather than visual fidelity.

**LoRA rank.** We study adaptation capacity by varying the LoRA rank under the fixed 50-video setting. We evaluate ranks $\{16, 32, 64, 128, 256\}$ and include full-parameter fine-tuning as an upper-capacity reference. Table 10b shows that higher ranks generally improve action accuracy while video quality remains largely stable. We use rank 16 in the main experiments as an efficient default, and this ablation confirms that our conclusions do not hinge on a specific rank choice.

## 5. Conclusion

We identify a key limitation in unsupervised latent action learning: *cross-context non-identifiability*. Inverse-dynamics objectives do not identify a global action basis, producing context-entangled latents that transfer poorly. We propose SeqΔ-REPA, a sequence-level objective that anchors latent actions to *action effects* measured as feature differences from a self-supervised video encoder, encouraging context-invariant semantics. Building on these latents, we introduce Olaf-World, a scalable latent-action world-modeling framework that improves zero-shot action transfer and enables data-efficient adaptation to new control spaces.

**Future work.** In robotics, effect-aligned latent actions could serve as transferable *skills* that bridge embodiments via an embodiment-specific action-to-skill adapter, *e.g.*, human→robot. We provide more discussion in Appendix E.

## Acknowledgements

This research is supported by the Ministry of Education, Singapore, under the Academic Research Fund Tier 1 (FY2023). Yuxin Jiang is supported by the A*STAR ACIS Scholarship.

## Impact Statement

This paper presents work whose goal is to advance the field of Machine Learning. There are many potential societal consequences of our work, none which we feel must be specifically highlighted here.

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

# Appendix

The document provides supplementary information not elaborated on in our main paper due to space constraints. It includes a formal analysis of cross-context non-identifiability (Section A), implementation details (Section B), evaluation protocols (Section C), additional results (Section D), and a discussion of limitations and future work (Section E). We also provide a project page (https://showlab.github.io/Olaf-World) with video visualizations that are essential for evaluating the temporal quality of the generated world models.

# A. Formal Analysis of Cross-Context Non-Identifiability

We formalize why standard local inverse-dynamics training signals do not, by themselves, identify a shared latent-action coordinate system across contexts. The key issue is a latent-coordinate symmetry (Locatello et al., 2019; Khemakhem et al., 2020; Wang et al., 2023b): the same transition predictions can be realized under different (context-dependent) reparameterizations of the latent codes.

## A.1. Setup

Let $c$ index a context (*e.g.*, viewpoint or scene). For each transition $(x_t, x_{t+1})$ from context $c$, a latent-action encoder $E$ and decoder $D$ are trained using the local prediction objective

$$\mathcal{L}_{\text{pred}}(E, D) = \mathbb{E}_c \, \mathbb{E}_{(x_t, x_{t+1}) \sim c} \Big[ \ell\big(x_{t+1}, \, D(x_t, E(x_t, x_{t+1}))\big) \Big], \qquad (6)$$

where $\ell(\cdot, \cdot)$ is any reconstruction/prediction loss (e.g., $\ell_2$).

## A.2. Proposition (context-dependent latent-coordinate symmetry)

**Proposition A.1.** *Fix any family of bijections $\{G_c : \mathbb{R}^{d_z} \to \mathbb{R}^{d_z}\}_c$ (one per context). Define a new encoder/decoder pair by*

$$E'(x_t, x_{t+1}) := G_c\big(E(x_t, x_{t+1})\big), \qquad (7)$$
$$D'(x_t, z) := D\big(x_t, \, G_c^{-1}(z)\big), \qquad (8)$$

*for transitions $(x_t, x_{t+1})$ from context c. Then $\mathcal{L}_{pred}(E', D') = \mathcal{L}_{pred}(E, D)$.*

*Proof.* For any transition in context $c$, substitute the definitions:

$$D'(x_t, E'(x_t, x_{t+1})) = D\Big(x_t, \, G_c^{-1}\big(G_c(E(x_t, x_{t+1}))\big)\Big) \qquad (9)$$
$$= D\big(x_t, E(x_t, x_{t+1})\big). \qquad (10)$$

Thus the prediction is identical for every sample, and taking expectations over $(x_t, x_{t+1})$ and $c$ leaves the loss unchanged. □

## A.3. Implication for cross-context transfer

Proposition A.1 implies that the latent representation is not anchored across contexts: different contexts can realize different latent coordinate systems while attaining the same training objective. Concretely, let $z^\star$ denote an abstract latent code producing a desired semantic effect, and suppose context $c_A$ represents it as $z_A := G_{c_A}(z^\star)$. Applying $z_A$ in a different context $c_B$ yields

$$D'(x_t^{(B)}, z_A) = D\big(x_t^{(B)}, G_{c_B}^{-1}(z_A)\big) = D\big(x_t^{(B)}, G_{c_B}^{-1} G_{c_A}(z^\star)\big), \qquad (11)$$

which generally differs from the intended effect unless $G_{c_B}^{-1} G_{c_A} \approx I$. Therefore, a code inferred in one context need not transfer as the "same action" in another.

### Remark (what changes with a $\beta$-VAE KL term?)

Many latent action models (Gao et al., 2025a; Garrido et al., 2026) include a $\beta$-VAE regularizer with isotropic prior, $\beta \, \text{KL}(q_\phi(z \mid \cdot) \, \| \, \mathcal{N}(0, I))$. The prior is rotationally invariant, but restricting $q_\phi$ to a factorized diagonal-Gaussian family breaks this continuous symmetry. For $q_\phi(z \mid \cdot) = \mathcal{N}(\mu(\cdot), \text{diag}(\sigma^2(\cdot)))$, the family is not closed under arbitrary orthogonal

rotations: if $z \sim \mathcal{N}(\mu, \Sigma_{\text{diag}})$ and $z' = Rz$ with orthogonal $R$, then $z'$ has covariance $R\Sigma_{\text{diag}}R^\top$, which is generally non-diagonal. Requiring $R\Sigma_{\text{diag}}R^\top$ to remain diagonal for all diagonal $\Sigma_{\text{diag}}$ forces $R$ to be a signed permutation matrix, up to degenerate isotropic cases. Thus, within this variational family, the remaining exact symmetries reduce to signed permutations of coordinates. Without an explicit cross-context constraint, these discrete symmetries can still vary with context, so latent directions remain non-identifiable across contexts and are not directly comparable for transfer.

## B. Implementation Details

### B.1. Latent Action Model

**Architecture.** Our LAM is implemented as a VAE-based video prediction framework consisting of a causal spatio-temporal encoder and a spatial-only decoder. Both the encoder and decoder have a Transformer architecture with 16 blocks, 1024 embedding dimensions, and 16 attention heads. The encoder applies causal masking to the temporal attention layers to prevent information leakage from future frames. Latent actions have dimension $d_z = 32$. We train on clips of length $T{=}16$ at resolution $272{\times}480$. For alignment, we use a projection head consisting of LayerNorm followed by a 3-layer MLP (Linear→SiLU→Linear→SiLU→Linear), projecting the pooled latent actions to the effect direction's dimension $D{=}1408$. As the frozen effect teacher, we use V-JEPA 2 ViT-Giant/16 (384) (Assran et al., 2025) pretrained on video data.

**Training.** We train with AdamW using learning rate $2.5{\times}10^{-5}$ and weight decay $10^{-2}$, with total batch size 32 on $8{\times}$H200 GPUs. We set $\beta{=}2{\times}10^{-4}$ for the KL term and $\lambda{=}0.02$ for the alignment loss. To preserve the fidelity of the effect trajectories extracted by V-JEPA 2, we disable color jitter during training. The model is trained for 100 epochs ($\sim$146k steps), taking $\sim$4.5 days.

### B.2. Olaf-World

**Architecture.** We build Olaf-World on the SkyReels I2V 1.3B DiT backbone (Chen et al., 2025a). We inject latent actions via a linear projection $32{\to}1536$ into the timestep embedding stream, with a learned gain $\gamma$ initialized to 2.0. For adaptation, we use LoRA with rank 16, applied to the `attn.{q,k,v,o}` and `ffn.{0,2}` linear layers in each DiT block.

**Training.** We pretrain the latent-action-conditioned video generator for $10k$ steps using AdamW with learning rate $5{\times}10^{-5}$ and weight decay $10^{-3}$. The training is distributed across $4{\times}$NVIDIA H200 GPUs with batch size 4 per device. For downstream adaptation, we fine-tune only LoRA parameters (rank $r{=}16$) with learning rate $1{\times}10^{-4}$ and zero weight decay.

## C. Evaluation Details

### C.1. Latent Space Diagnostics

#### C.1.1. CROSS-CONTEXT LINEAR PROBING

**Probe training.** We train a single linear classifier on top of frozen latent actions $z_t$. We optimize with SGD (momentum 0.9, weight decay $10^{-6}$) for 12 epochs using a StepLR schedule. To handle class imbalance, we use focal loss with $\gamma{=}2$. For each training domain, we select the checkpoint that achieves the highest in-domain validation Macro-F1.

**Cross-domain evaluation.** We evaluate the selected probe zero-shot on the other domain and report Macro-F1.

#### C.1.2. CROSS-CONTEXT ACTION CONSISTENCY

**Prototype construction.** For each domain $d \in \{\text{1ST-P}, \text{3RD-P}\}$, we sample clips and infer per-step latent actions with the pretrained LAM. For each action class $c$, we collect the corresponding latents $\mathcal{Z}_c$ and compute the prototype (class centroid):

$$\mathbf{p}_c = \frac{1}{|\mathcal{Z}_c|} \sum_{\mathbf{z} \in \mathcal{Z}_c} \mathbf{z}. \tag{12}$$

**Cross-domain similarity matrix.** Given the two prototype matrices $P^{(\text{1ST-P})} \in \mathbb{R}^{C \times d_z}$ and $P^{(\text{3RD-P})} \in \mathbb{R}^{C \times d_z}$ ($C{=}8$ actions), we preprocess each by $\ell_2$-normalizing. We then compute the cosine similarity heatmap:

$$S_{ij} = \cos\left(\mathbf{p}_i^{(\text{1ST-P})}, \mathbf{p}_j^{(\text{3RD-P})}\right) = \frac{\langle \mathbf{p}_i^{(\text{1ST-P})}, \mathbf{p}_j^{(\text{3RD-P})} \rangle}{\|\mathbf{p}_i^{(\text{1ST-P})}\|_2 \|\mathbf{p}_j^{(\text{3RD-P})}\|_2}. \tag{13}$$

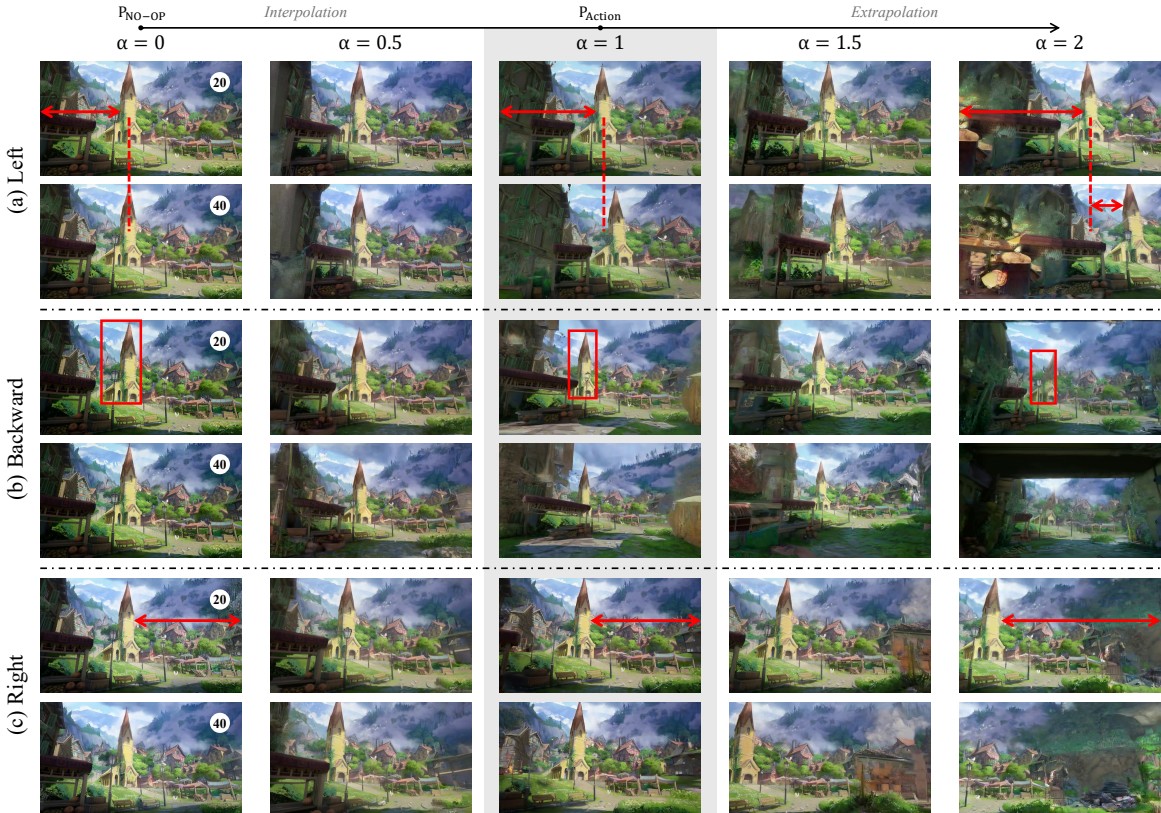

*Figure 11.* **Latent action space interpolation and extrapolation.** We interpolate between the no-operation prototype $P_{\text{NO-OP}}$ and an action prototype $P_{\text{ACTION}}$ in the latent action space using $\alpha \in \{0, 0.5, 1, 1.5, 2\}$ (columns). Rows correspond to actions (a) left, (b) backward, and (c) right. For each $\alpha$, the same interpolated latent is repeated across all action steps and applied to the same initial image. We show generated frames at timesteps 20 and 40.

Rows correspond to 1ST-P prototypes and columns to 3RD-P prototypes.

### C.2. World model

**Visual quality.** We evaluate visual quality using selected dimensions from VBench (Huang et al., 2024), including *Imaging Quality* and *Temporal Consistency*.

**Action accuracy via relative pose error.** Following (Hong et al., 2025), we adopt a behavioral protocol to evaluate controllability. Given a fixed action sequence, the model generates the video. We then reconstruct the induced camera trajectories from both the ground-truth (GT) and generated videos using ViPE (Huang et al., 2025b), which estimates per-frame camera poses. We align the generated trajectory to the GT trajectory using a Sim(3) Umeyama alignment to remove scale and coordinate-frame differences. Finally, we compute *Relative Pose Error* (RPE) between the GT and aligned generated trajectories, reporting (i) **RPE-trans**, the translation error magnitude, and (ii) **RPE-rot**, the rotation error angle. Lower values indicate better agreement with the intended camera motion.

**Novel-scene dataset construction.** Since the MIND (Ye et al., 2026) training distribution primarily consists of near-photorealistic 3D game renderings, we curate an OOD novel-scene evaluation set of 50 initial frames to test robustness under large appearance shifts. The set spans diverse visual domains, including photorealistic scenes (Huang et al., 2024) and stylized images such as anime and oil paintings (Jiang et al., 2023; World Labs, 2025). For evaluation, all models are conditioned on these novel-scene frames and driven by the same sequence of actions used for in-domain testing.

## D. Additional Results

We provide additional qualitative examples for zero-shot action-sequence transfer, data-efficient adaptation, and generalization to novel scenes. These results further confirm the robustness and superior performance of our method compared to

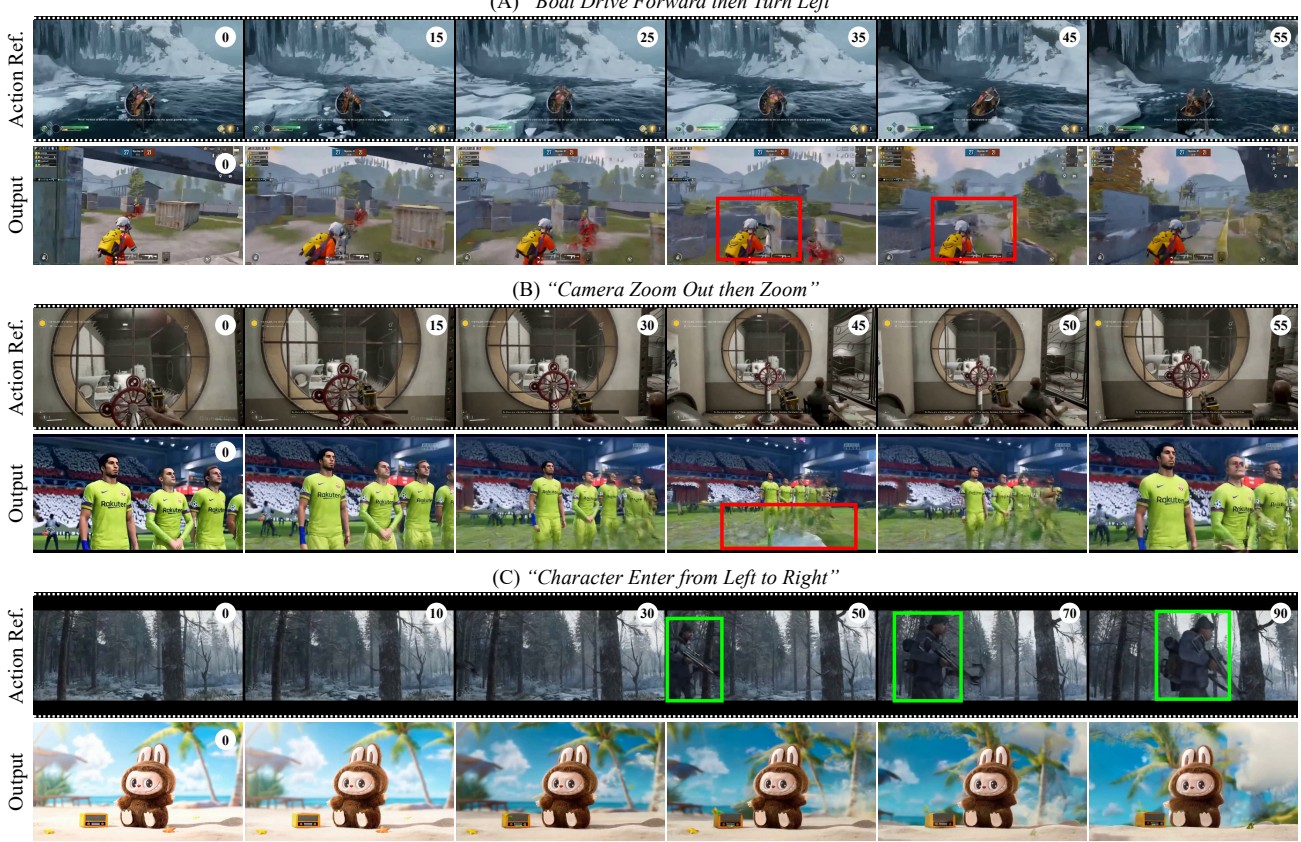

Figure 12. **Failure cases generated by Olaf-World.**

baselines. Due to the inherent difficulty of conveying dynamic video generation quality through sparsely sampled frames, we refer readers to the supplementary project page for the corresponding videos: https://showlab.github.io/Olaf-World.

**Latent Action Interpolation.** To further probe the local geometry of the learned latent action space, we conduct a simple qualitative linear interpolation experiment. Using the prototypes defined in Section C.1.2, for each domain $d$ and action $a$, we interpolate between the NO-OP prototype and the target action prototype:

$$\tilde{\mathbf{z}}_{d,a}^{(\alpha)} = (1 - \alpha)\mathbf{p}_{d,\text{NO-OP}} + \alpha\mathbf{p}_{d,a}, \tag{14}$$

where $\alpha$ controls the action strength. When $0 < \alpha < 1$, Eq. (14) performs interpolation, while $\alpha > 1$ corresponds to extrapolation. For each $\alpha$, we repeat the same interpolated latent across all action steps and use it to condition video generation from the same initial image. As shown in Figure 11, increasing $\alpha$ produces progressively stronger motion in the intended direction under the same repeated action signal, suggesting a reasonably coherent local geometry of the learned latent action space.

**Failure cases.** Figure 12 shows three representative failure cases: **(A) Control–physics mismatch.** When a transferred action would cause collisions in the target scene (*e.g.*, drive forward then turn left), the model may hallucinate scene changes to remove or alter the obstacles to avoid collisions, thus preserving the intended motion. **(B) Degraded completion under large reveal.** Actions such as zooming out require synthesizing a large amount of newly visible content. In these cases, extended parts of the video (*e.g.*, players' legs) may appear blurry or inconsistent. **(C) Ambiguous realization for event-driven actions.** For actions that imply an event (*e.g.*, a new character entering), the identity of the entering entity is not specified under cross-context transfer. In our example, the model realizes the control as background/camera drift while keeping existing subjects consistent, which is a plausible relative-motion interpretation, but not the same event semantics. We leave richer event-level transfer (*e.g.*, controlled object entry) to future work.

# E. Limitations and Future Work

We outline several promising directions that could further strengthen transferable latent actions and action-conditioned world modeling.

## E.1. Effect-aligned latent actions

**Objectives and effect targets.** We use a simple and effective cosine alignment between latent actions and effect directions defined by feature differences from a frozen video encoder. Exploring alternative effect targets and alignment formulations is a natural next step and may further improve robustness across diverse contexts and the structure of the learned latent action space.

**Hierarchical latents (skills).** Our current latent actions are step-level (one latent per frame at 16 FPS). Learning a hierarchy of latent actions, where short-horizon controls compose into longer-horizon "skills", may improve long rollouts, enable multi-rate control, and provide a cleaner interface for downstream decision-making (Hafner et al., 2022; Gumbsch et al., 2024).

**Toward physics-rule transfer.** A natural next step is to augment effect-aligned latent actions with physics-grounded constraints so that transferred trajectories remain visually faithful and physically plausible. Recent work shows video generators can be post-trained with *verifiable* kinematic or collision-consistency rewards (*e.g.*, Newtonian acceleration for falling objects, collision rules) to improve physical behavior (Le et al., 2025; Zhang et al., 2026). A further step is to extend action-conditioned transfer to **contact-rich interactions**, which require continuous contacts between multiple objects, moving beyond navigation toward complex manipulation.

**Multi-entity dynamics and factorized control.** Seq$\Delta$-REPA currently summarizes the observed change with a single effect signal, which can mix different sources of control, camera/ego motion, controllable agent motion, other agent behavior, and environment-driven events. Factorizing effects (ego vs. others vs. environment) and learning entity-specific latent control could improve interpretability and enable richer multi-entity controllable world modeling.

## E.2. Latent actions for planning and reasoning

**Planning and sampling in latent-action space.** In this work, latent actions are used for transfer and as a control interface via an adapter. A key next step is to plan directly on latent action sequences using the world model for imagination-based search or trajectory optimization (Rybkin et al., 2019; Hafner et al., 2023; Wu et al., 2023; Hafner et al., 2022; Yu et al., 2020).

**From frame-level "visual CoT" to latent-action traces.** Recent work shows that large video models can exhibit emergent zero-shot capabilities (Wiedemer et al., 2025), and video generation work has begun to use *visual chain-of-thought—e.g.*, sparse keyframes, intermediate "thought" prompts, or storyboard plans—as guidance to improve long-horizon coherence and controllability (Rotstein et al., 2025; Huang et al., 2025d; Xiao et al., 2025a; Huang et al., 2025a; Li et al., 2025). An intriguing direction is to treat latent-action sequences as compact *traces of dynamics* that are cheaper and less redundant than dense frame-level visual CoT, and to study how such traces can support evaluation, editing, and higher-level reasoning about actions and events.

