# OpenReview forum: "Olaf-World: Orienting Latent Actions for Video World Modeling"
_ICML.cc/2026/Conference — ICML 2026 regular_

### Official Review · Reviewer_JN98 · 2026-03-08

**Soundness:** 3
**Presentation:** 3
**Significance:** 3
**Originality:** 3
**Overall Recommendation:** 4
**Confidence:** 4

**Summary:**

This paper studies learning transferable latent action representations from passive videos for video world modeling. The authors argue that standard latent action objectives based on local reconstruction may produce context-dependent latent spaces. To address this, the paper proposes SeqΔ-REPA, which aligns latent action sequences with an effect direction derived from feature differences of a frozen pretrained video encoder. The learned latent actions are then used as a control interface to pretrain an action-conditioned video world model. Experiments evaluate cross-domain consistency, zero-shot action transfer, and adaptation to environment-specific action spaces.

**Compliance With Llm Reviewing Policy:**

Affirmed.

**Key Questions For Authors:**

1.	The proposed SeqΔ-REPA aligns latent actions to an effect direction computed as the feature difference from a frozen video encoder. However, it is unclear why such feature differences should reliably correspond to semantic control effects. Pretrained video representations may entangle scene appearance, camera motion, or other dynamics unrelated to agent control, and the same action (e.g., moving forward) could produce very different feature changes in visually distinct environments. Why can a pretrained video encoder provide a stable and reliable reference for action semantics across contexts?
2.	Another question concerns the choice of reference space for defining the effect direction. The paper uses feature differences from a pretrained video encoder, while the world model itself already learns latent state representations that capture environment dynamics. Why not define the effect direction directly in the learned latent state space (e.g., using state differences) instead of relying on external visual features?
3.	Appendix A shows that latent coordinates learned under local reconstruction objectives are not identifiable across contexts. However, it is unclear why such ambiguity necessarily leads to poor controllability or transfer. In many world-model approaches [1,2], latent representations are also non-identifiable up to coordinate transformations yet still support effective planning and control. Why is this ambiguity particularly problematic in this setting?
4.	SeqΔ-REPA aggregates a sequence of latent actions through simple averaging before aligning it with the effect direction (Eq.(2) and (3)). Since this operation ignores temporal ordering, different action sequences could produce similar averaged representations. Could the authors clarify why this preserves meaningful control semantics?
5.	Some qualitative comparisons in Fig.6 (e.g., case C) show only minor visual differences between methods, making the improvement somewhat difficult to assess. It would be helpful to clarify whether the transferred action sequences are identical across methods and whether the reference clips have comparable sequence length and complexity.
6.	The method relies on a frozen pretrained video encoder as the reference signal. How sensitive are the results to the choice of encoder? Similarly, how sensitive is the performance to the weighting parameter λ? Are there ablations evaluating these factors?

Reference: [1] Bar, Amir, et al. "Navigation world models." CVPR. 2025.
[2] Hafner, Danijar, et al. "Mastering diverse domains through world models." arXiv preprint arXiv:2301.04104 (2023).

**Limitations:**

yes

**Strengths And Weaknesses:**

Strengths:

1.	The paper studies the problem of learning transferable latent action representations from passive videos, which is relevant for video world models and controllable video generation.

2.	The proposed SeqΔ-REPA objective is simple and easy to integrate into existing latent action learning frameworks.

3.	The experimental evaluation is relatively comprehensive, including cross-domain probing, zero-shot transfer, and adaptation experiments.

Weaknesses:

1.	The method relies on feature differences from a frozen pretrained video encoder as the reference signal, but the paper does not provide strong justification that this signal reliably corresponds to semantic action effects.

2.	Some qualitative comparisons (e.g., Fig.6C) show only minor visual differences, and the sensitivity to design choices such as the reference encoder or the loss weight λ is not analyzed.

---

> ### Author Rebuttal · Authors · 2026-03-31
>
> We thank Reviewer JN98 for the thoughtful review and positive evaluation. Below we address each point and will include all results and discussions in the revision.
>
> > _1. [...] why such feature differences should reliably correspond to semantic control effects [...] Why can a pretrained video encoder provide a stable and reliable reference for action semantics across contexts?_
>
> - First, Δ suppresses static scene content (appearance, lighting, layout) that is approximately constant within a short clip, whereas the action effect is expressed through **change over time**. Our ablation (Table 4) validates this: removing Δ degrades cross-domain transfer (1st→3rd: 62.5→52.9, 3rd→1st: 59.0→48.2).
>
> - Second, video SSL encoders are trained on large-scale video to learn spatiotemporal representations across diverse scenes. If feature differences were dominated by context-dependent appearance rather than action semantics, then video and image encoders should perform similarly. Our encoder ablation (Q6 below) shows the opposite: video encoders dramatically outperform image-only DINOv3, suggesting that video pretraining provides a more temporally structured effect space for cross-context alignment.
>
> > _2. [...] Why not define the effect direction directly in the learned latent state space [...] instead of relying on external visual features?_
>
> We clarify that the "world model" in this paper refers to an action-conditioned video diffusion model, not a latent state-space model with an explicit learned state. Our goal is to learn a transferable latent action interface from passive video.
>
> For training stability, we use a two-stage pipeline: (1) latent action learning from pixels, then (2) latent-action-conditioned video world modeling. Thus, when learning latent actions, there is no world-model latent state available to define the effect direction.
>
> > _3. [...] Why is this ambiguity particularly problematic in this setting?_
>
> The non-identifiability discussed in Appendix A concerns the learned latent action space, not latent (world) state representations.
>
> - NWM [1] and DreamerV3 [2] use **predefined explicit action spaces**, not learned latent actions. Their latent states may be non-identifiable, but their action interface is fixed, so controllability is unaffected.
>
> - Our setting is different: **the actions themselves are learned from passive video** and should serve as a shared control interface across contexts, which is precisely where non-identifiability becomes problematic.
>
> > _4. SeqΔ-REPA aggregates a sequence of latent actions through simple averaging before aligning it with the effect direction. Since this operation ignores temporal ordering [...] why this preserves meaningful control semantics?_
>
> The temporal ordering is preserved by the reconstruction loss (standard VAE objective), which requires each individual $z_i$ to predict $x_{i+1}$ from $x_i$ in sequence. This ensures each $z_i$ remains individually meaningful and correctly ordered. SeqΔ-REPA is a regularizer on the aggregate effect direction, not the sole training signal. Reconstruction handles temporal ordering; alignment handles semantic grounding.
>
> > _5. [...] Fig.6 (e.g., case C) show only minor visual differences between methods [...] whether the transferred action sequences are identical across methods [...]_
>
> - All methods use the same reference video clip. Each method extracts its own latent action sequence from that clip then conditions the video world model.
>
> - In Fig. 6C, the reference contains a large rightward and downward camera motion that brings the ground into view. Our method reproduces this faithfully, while the baseline generates visually plausible frames but with much smaller movement, failing to match the reference video's effect. We refer the reviewer to the supplementary videos where these differences are clearly visible. We will add figure annotations in the revision.
>
> > _6.  [...] How sensitive are the results to the choice of encoder [...] weighting parameter λ?_
>
> Following the reviewer's suggestion, we run additional ablations (same setup as Sec. 4.2.1; 10-epoch training due to the rebuttal timeline).
>
> - Encoder sensitivity
>
>     | ref encoder            | 1st→1st | 1st→3rd | 3rd→3rd | 3rd→1st |
>     |--|--:|--:|--:|--:|
>     | V-JEPA2                 | 0.704   | 0.297   | 0.585   | 0.470   |
>     | VideoMAEv2              | 0.663   | 0.470   | 0.662   | 0.538   |
>     | DINOv3                  | 0.233   | 0.178   | 0.287   | 0.191   |
>     | Unaligned      | 0.185   | 0.138   | 0.211   | 0.159   |
>
>     Video encoders ≫ image-only (DINOv3), indicating that temporal modeling in the reference encoder is key.
>
> - λ sensitivity
>
>     | λ         | 1st→1st | 1st→3rd | 3rd→3rd | 3rd→1st |
>     |--|--:|--:|--:|--:|
>     | 0.02                 | 0.7044   | 0.2972   | 0.5845   | 0.4704   |
>     | 0.002              | 0.5959   | 0.2929  | 0.3994   | 0.3782   |
>
>     Reducing λ lowers performance but remains well above the unaligned baseline.

---

> > ### Author Rebuttal · Reviewer_JN98 · 2026-04-03
> >
> > My issues have been addressed by the authors.
> > I will keep my original positive score. Thank you for the response.

---

### Official Review · Reviewer_Uu9D · 2026-03-11

**Soundness:** 2
**Presentation:** 2
**Significance:** 3
**Originality:** 3
**Overall Recommendation:** 4
**Confidence:** 3

**Summary:**

The paper addresses one of the common challenges in building action-conditioned video world models from unannotated data. The learned actions often don't transfer well across contexts. The authors claim that conventional reconstruction-guided latent actions models often entangle scene-specific cues with action semantics, which leads to poor transferability. To solve this, the paper proposes Seq$\Delta$-REPA that aligns predicted latent actions with temporal differences in the latent spcae of a pre-trained self-supervised video model. Building on this, the authors introduce Olaf-World, a world model that improves zero-shot action transfer and data-efficient adaptiation to ground truth action spaces over the prior work.

**Compliance With Llm Reviewing Policy:**

Affirmed.

**Final Justification:**

The rebuttal addressed my concerns, therefore I update my rating accordingly. In particular, the ablation of the reference encoder is quite insightful and indicative of the methods effectiveness.

**Key Questions For Authors:**

1) Isn't equation (2) just collapsing to $\frac{1}{K}(s_K - s_0)$. How is this then averaged over time?

**Limitations:**

yes

**Strengths And Weaknesses:**

### Strengths
1) The paper addresses an important issue of non-transferability of latent actions and provides a good intuition of why the local reconstruction-based objectives may fail to learn shared action semantics across contexts.
2) Seq$\Delta$-REPA is a simple and novel regularization term that distills motion priors from pre-trained video encoders.
3) The method shows improvements in cross-domain linear probing and action consistency heatmaps compared to the prior AdaWorld baseline.

### Weaknesses
1) The experiments are primarily focused on navigation and camera control in Unreal Engine environments. It is unclear how well this effect-alignment generalizes to more complex, manipulation or multi-agent dynamics.
2) The performance of Seq$\Delta$-REPA is heavily dependent on the quality of the frozen video encoder (V-JEPA 2 in this case). The paper lacks a sensitivity analysis on how different teacher models affect the learned latent space.
3) The presentation of the method can be improved (see questions for authors). There are several inconsistencies between the text and the figures. E.g. the figures depict video MAE as the pre-trained video encoder, while the text refers to V-JEPA 2. $\Delta y_0$, $\Delta y_1$ in Figure 3 is never mentioned in the text.

---

> ### Author Rebuttal · Authors · 2026-03-31
>
> We thank Reviewer Uu9D for the constructive feedback and recognizing the importance of the problem, the novelty of Seq$\Delta$-REPA, and the improvements over AdaWorld. We address each concern below and will incorporate these clarifications in the revision.
>
> > _1. The experiments are primarily focused on navigation and camera control in Unreal Engine environments. It is unclear how well this effect-alignment generalizes to more complex, manipulation or multi-agent dynamics._
>
> - We agree that extending to manipulation and multi-agent dynamics is an important next step. We focus on movement and camera control because latent action learning remains challenging even in this basic but important setting, which is the primary control interface in game and simulation environments (e.g., Genie 3).
> - We also please refer the reviewer to the supplementary videos, which show several action transfer results beyond navigation, including flying, shooting, attacking, and  despawning. Since Seq$\Delta$-REPA is architecture and domain agnostic (a single alignment term with no domain-specific assumptions), we expect the same principle to extend to richer settings. We discuss this limitation and future direction in Appendix E.
>
> > _2. The performance of Seq$\Delta$-REPA is heavily dependent on the quality of the frozen video encoder (V-JEPA 2 in this case). The paper lacks a sensitivity analysis on how different teacher models affect the learned latent space._
>
> Following the reviewer's suggestion, we run a visual encoder ablation using the same in-/cross-domain linear probing setting as Sec. 4.2.1, and report Macro-F1 below. Due to the rebuttal timeline, these numbers are from 10-epoch latent-action training.
>
> | reference encoder            | 1st→1st | 1st→3rd | 3rd→3rd | 3rd→1st |
> |-----------------------------|--------:|--------:|--------:|--------:|
> | **V-JEPA2**                 | 0.7044   | 0.2972   | 0.5845   | 0.4704   |
> | **VideoMAEv2**              | 0.6626   | 0.4697   | 0.6616   | 0.5377   |
> | **DINOv3**                  | 0.2334   | 0.1782   | 0.2869   | 0.1910   |
> | **Unaligned baseline**      | 0.1849   | 0.1382   | 0.2105   | 0.1593   |
>
> We can see video encoders (V-JEPA2, VideoMAEv2) $\gg$  image-only encoder (DINOv3), and different video SSL encoders are both effective. The results indicate that Seq$\Delta$-REPA **does not depend on a specific encoder's feature quality**; instead, **temporal modeling capability is the key factor**.
>
>
> > _3. [...] There are several inconsistencies between the text and the figures. [...]_
>
> We thank the reviewer for the careful reading. We will fix these in the revision. Specifically:
> - Figure 3 encoder label: we will update the label to ''pretrained video encoder''.
> - $\Delta$y in Figure 3: We will add explicit definitions accompanying Figure 3.
>
>
> > _4. Isn't equation (2) just collapsing to ${1 \over K}(s_K - s_0)$​. How is this then averaged over time?_
>
> Yes. We formulated Eq.2 as an average of per-step differences rather than the equivalent endpoint form ${1 \over K}(s_K - s_0)$ to emphasize the design principle: the alignment target measures the accumulated local effects over the clip, connecting naturally to the per-step reconstruction objective and making the intuition easier to follow.
> Our implementation is consistent with this, and we will add the endpoint form explicitly in the revision for clarity.

---

> > ### Author Rebuttal · Reviewer_Uu9D · 2026-04-02
> >
> > Thank you for the rebuttal. The authors have addressed all my concerns.

---

### Official Review · Reviewer_ZuZd · 2026-03-13

**Soundness:** 3
**Presentation:** 4
**Significance:** 3
**Originality:** 3
**Overall Recommendation:** 5
**Confidence:** 4

**Summary:**

The paper brings up the problem of context-dependence to low-level visual cues when inferring latent actions from passive videos. They point to problems such as the same latent action might hold different meanings in different contexts. Or, conversely, the same action may be represented by different latent actions in different contexts.

Their solution is to add a new loss term to the conventional latent action learning using VAE framework. The conventional framework adopts a VAE to encode a stack of historical frames into a latent action z. The latent action z can then be used to predict the final frame in the stack given its history. In addition to standard VAE loss, a new term is introduced. For this, first, all the frames are encoded using a pre-trained image encoding backbone (V-JEPA2). Next, the per-step differences in these encodings are aggregated across the clip to get \tau*. Finally, a cosine similarity is computed between aggregated z’s from the VAE portion and the \tau*.

They then make a pretrained image-to-video model controllable using the aforementioned action latents. For this, they take a passive video dataset, infer all the latent actions, and the fine-tune the pre-trained image-video model to accept the latent actions as input – thus making the world model action controlled.

In experiments, they extensively test several aspects: (1) cross-context transfer of linear probes that classify actions; (2) cross-context similarity in prototypes/average latent actions corresponding to identical actions in different contexts; (3) zero-shot action transfer i.e., take the latent action sequence from video A to drive video B and see how well the generated video B matches the true expectation. (4) Adapt the world model to known action labels using a small number of action-labeled examples. (5) OOD contexts. Across the board, they show good benefits of their method. Ablations are also useful.

**Compliance With Llm Reviewing Policy:**

Affirmed.

**Final Justification:**

The response also addresses my other concerns and I continue to support acceptance as before.

**Key Questions For Authors:**

- Isn’t Eq. 2 simplifiable to $(s_K - s_0)/K$? Thus, it doesn’t need per-frame differences to compute? Also, why perform these aggregations across the clip length? Why not have this constraint loss per pair of frames?
- Do the videos reveal actions on the screen or is that merely for visualization in Figure 7?

**Limitations:**

yes

**Strengths And Weaknesses:**

Strengths
- Addresses an important problem of inferring causality from passive videos, specifically, inferring actions from unlabelled videos.
- Identifies an important shortcoming of existing approaches. Tackles it via a novel and interesting approach
- Shows strong results across the experiments.
- Clear writing, experiment design, and diagrams.

Weaknesses
- Relies on a pre-trained V-JEPA2 to obtain the benefits that they see. It could help if they could explain why this specific encoder is chosen. It appears that this is a black-box unexplained element in their entire pipeline – despite having remarkable results. In this regard it may help to test other image encoding backbones. That may shed insights for practitioners in what to look for when picking an image encoder.
- The considered action space is only the tip of the ice-berg when it comes to extracting causal scene factors e.g., what if there are multiple actors in the scene. However, those may rightly lie outside the scope of the current work. In that regard, it may be asked how far do the authors foresee that the pre-trained V-JEPA2 style backbones can remain effective in the long run.

---

> ### Author Rebuttal · Authors · 2026-03-31
>
> We sincerely thank Reviewer ZuZd for the thorough reading, insightful comments and positive assessment. We are encouraged that the reviewer finds the problem important, the approach novel, the results strong, and the presentation clear. We address each question below and will include all results and discussions in the revision.
>
> > _1. [...] . It could help if they could explain why this specific VJEPA encoder is chosen. [..] In this regard it may help to test other image encoding backbones. [...]_
>
> We chose V-JEPA2 because its self-supervised video features are semantically meaningful and temporally consistent, making it a natural reference for measuring short-horizon effect directions.
>
> Following the reviewer's suggestion, we run a visual encoder ablation using the same in-/cross-domain linear probing setting as Sec. 4.2.1, and report Macro-F1 below. Due to the rebuttal time limitation, these numbers are from 10-epoch latent-action training.
> | reference encoder            | 1st→1st | 1st→3rd | 3rd→3rd | 3rd→1st |
> |-----------------------------|--------:|--------:|--------:|--------:|
> | **V-JEPA2**                 | 0.7044   | 0.2972   | 0.5845   | 0.4704   |
> | **VideoMAEv2**              | 0.6626   | 0.4697   | 0.6616   | 0.5377   |
> | **DINOv3**                  | 0.2334   | 0.1782   | 0.2869   | 0.1910   |
> | **Unaligned baseline**      | 0.1849   | 0.1382   | 0.2105   | 0.1593   |
>
> These results suggest three takeaways.
> - (i) All encoders improve over the baseline, confirming that the benefit of Seq$\Delta$-REPA is not specific to a single backbone.
> - (ii) Video encoders (V-JEPA2, VideoMAEv2) >> image-only encoder (DINOv3), suggesting that temporal modeling in the reference encoder is the key factor.
> - (iii) V-JEPA2 (predictive semantic objective) and VideoMAEv2 (reconstruction-based objective) are both effective, suggesting the most important practical criterion is a frozen representation space with strong temporal consistency and video-level dynamics awareness. We will include this discussion in the revision.
>
> > _2. [...] e.g., what if there are multiple actors in the scene. [...] how far do the authors foresee that the pre-trained V-JEPA2 style backbones can remain effective in the long run._
>
> We agree with the reviewer and discuss multi-entity dynamics and factorized control as a limitation and future direction in Appendix E. More broadly, we do **not** view V-JEPA2 style backbones as a complete long-term solution in themselves; rather, we view them as a powerful temporal reference space whose usefulness persists as self-supervised video representations improve.  In other words, **Seq$\Delta$-REPA is complementary to advances in video SSL and should directly benefit from stronger future representations.**
>
> For multi-entity scenes, newer models such as V-JEPA2.1, with denser and more spatially structured features, may enable more spatially factorized alignment, extending Seq$\Delta$-REPA from ''what action'' toward ''who acted where''. We leave this to future work. The current paper focuses on establishing the temporal effect alignment principle for improving cross-context consistency.
>
> > _3. Isn't Eq. 2 simplifiable to $(s_K - s_0)/K$? Thus, it doesn’t need per-frame differences to compute? Also, why perform these aggregations across the clip length? Why not have this constraint loss per pair of frames?_
>
> Yes. Eq.2 simplifies to $(s_K - s_0)/K$. We formulated it as an average of per-step differences to emphasize the design principle: the alignment target measures the accumulated local effects over the clip, connecting naturally to the per-step reconstruction objective and making the intuition easier to follow.
> Our implementation bases on this, and we will add the endpoint form in the revision for clarity.
>
> Why clip-level rather than per-pair? At 16 fps, the visual change between two consecutive frames is often subtle, so the resulting $\Delta$s may be a weak signal in the video encoder's feature space. Aggregating over a short horizon (16 frames, ~1 second) provides a more stable and semantically meaningful effect direction.
>
> > _4. Do the videos reveal actions on the screen or is that merely for visualization in Figure 7?_
>
> The keyboard indicators shown in Figure 7 are **only visualization overlays** added for presentation so that readers can easily associate each transition with the intended action. They are not part of the visual input and not part of the generated scene content. We will clarify this in the caption to avoid ambiguity.

---

> > ### Author Rebuttal · Reviewer_ZuZd · 2026-04-01
> >
> > Thank you for the great rebuttal. The backbone ablation is very insightful especially the distinction between the image-only and the video-centric backbones. The response also addresses my other concerns and I continue to support acceptance as before.

---

### Decision · Program_Chairs · 2026-04-30

**Decision:**

Accept (regular)

**Comment:**

This paper proposes a pretraining objective for improving the generalization of action-condition video models. This paper received several positive reviews, and after the rebuttal the reviewers acknowledge that all their concerns have been addressed. Reviewers highlight the importance of the problem, the strong results, and the proposed method, which is straightforward to integrate. Initial weaknesses focused on the paper's focus on VjEPA and design choices, which have since been addressed. In particular an analysis using different backbones/teachers was provided, which makes it clear the method generalizes.